# Burnout in Professional Psychotherapists: Relationships with Self-Compassion, Work–Life Balance, and Telepressure

**DOI:** 10.3390/ijerph18105308

**Published:** 2021-05-17

**Authors:** Yasuhiro Kotera, Robert Maxwell-Jones, Ann-Marie Edwards, Natalie Knutton

**Affiliations:** Human Sciences Research Centre, University of Derby, Derby DE22 1GB, UK; R.Maxwell-Jones@derby.ac.uk (R.M.-J.); annm.edwards@icloud.com (A.-M.E.); N.Knutton1@unimail.derby.ac.uk (N.K.)

**Keywords:** self-compassion, telepressure, work–life balance, burnout, depersonalisation, emotional exhaustion

## Abstract

Though negative impacts of COVID-19 on occupational mental health have been reported, the mental health of psychotherapists has not been evaluated in depth. As this occupational group treats ever-increasing mental health problems, it is essential to appraise key factors for their mental health. Accordingly, this study aimed to explore burnout of professional psychotherapists. A total of 110 participants completed self-report measures regarding burnout, self-compassion, work–life balance and telepressure. Correlation, regression and moderation analyses were conducted. Both of the burnout components—emotional exhaustion and depersonalisation—were positively associated with weekly working hours and telepressure, and negatively associated with age, self-compassion and work–life balance. Weekly working hours and work–life balance were significant predictors of emotional exhaustion and depersonalisation. Lastly, self-compassion partially mediated the relationship between work–life balance and emotional exhaustion but did not mediate the relationship between work–life balance and depersonalisation. The findings suggest that maintaining high work–life balance is particularly important for the mental health of psychotherapists, protecting them from burnout. Moreover, self-compassion needs to be cultivated to mitigate emotional exhaustion. Mental health care for this occupational group needs to be implemented to achieve sustainable mental health care for workers and the public.

## 1. Introduction

### 1.1. High Rates of Burnout in Professional Psychotherapists

The impacts of COVID-19 and the changes it has brought to general mental health levels is attracting different levels of global research. One area that appears to have been overlooked is the impact on the profession that is providing mental health for others [1,2]. The existing studies report increasing levels of poor mental health in the healthcare workforce [3,4]. With up to 40% of healthcare workers reporting depressive symptoms and 45% reporting severe anxiety symptoms lasting up to three years post pandemic (e.g., SARS) [5,6], more focus needs to be given to mental health professionals supporting those in need. Despite these concerning findings the mental health in this workforce remains to be evaluated. One reason for this could be the apparent stigma attached to mental health amongst healthcare providers [7]. This should be a motivation to fully appraise the levels of mental health and burnout in members of this profession field such as psychotherapists. It is also especially important with the increase in mental health support required post COVID as burnout is considered common in this profession [8,9].

### 1.2. Self-Compassion

Recent studies have found correlations between self-compassion and wellbeing, especially in the wake of COVID-19 where mental health is being impacted by continued uncertainty [10,11]. Self-compassion has been long documented as helping improve mental health [12]. It comprises of three main elements; self-kindness vs. self-judgement, common humanity vs. isolation and mindfulness vs. overidentification [13]. It encourages a person to be kind to themselves when they may not be feeling at their best—not berate themselves and worsen their mood (self-kindness). As well as this, self-compassion shows the importance of the shared experience when it comes to emotion. If someone is in pain, they are not the only one to feel pain; it is part of the human experience (common humanity). This leads to the third element, which is about recognising suffering as it is non-judgementally (mindfulness), instead of denying or fighting.

Individuals with a high level of self-compassion accept themselves for who they are and take care of themselves, which helps them in times of hardship making them more resilient to mental distress [3,14]. Self-compassion has been identified as a key construct to protect healthcare workers’ mental health during COVID-19 [15,16].

### 1.3. Burnout and Work–Life Balance

Burnout is a syndrome conceptualised as resulting from chronic workplace stress that has not been successfully managed. It is characterised by three dimensions: (a) feelings of energy depletion or exhaustion, (b) increased mental distance from one’s job, or feelings of negativism or cynicism related to one’s job, and (c) reduced professional efficacy [17,18].

Maslach and Leiter (2016) found that burnout research has brought the emergence of three demands-resources based models of burnout. The Job Demands–Resources (JD-R) model [17] focusses on the concept that burnout occurs when people face constant work demands but do not have resources available to them to resolve or reduce these demands. Similarly, the Conservation of Resources (CoR) model [19] posits that, when available resources are constantly threatened, work motivation is reduced leading to poor mental health. A new variation of the models above is the Areas of Worklife (AW) model [20], which considers a more person-centred approach to the areas of stress, which contribute to burnout, including workload. These models highlight the importance of work–life balance to burnout.

The current pandemic is impacting global mental health and increasing the numbers of people needing support [21,22,23]. As demand increases so will the workload for psychotherapists. Increased workload (e.g., longer working hours, limited resources) and associated pressure (e.g., threat of exposure to the virus, lack of updated information) impact work–life balance, which is a contributing factor to burnout [18,24]. These models and findings suggest psychotherapists are at risk of burnout, however research evaluating this mental health construct in this workforce has been thin to date.

### 1.4. Telepressure

Telepressure is a fixation with checking and responding to messages quickly [25]. Within the current pandemic the working lives of many have changed dramatically. A significant change for many has been with the movement of work from offices to home. In April 2020, 46.6% of people in the UK worked from home. A total of 86% of those people worked from home as a result of COVID-19 [26]. One of the occupational groups impacted substantially by new working arrangements are psychotherapists. Restrictions at different times have meant that face-to-face appointments with clients have not always been allowed and this has seen an increase in video and telephone counselling sessions [27]. With the change in mode for delivery of psychotherapy, Shklarski, Abrahams and Bakst (2021) also found issues such as ‘zoom fatigue’ being reported. It was also found that in some cases psychotherapists needed to support their clients more to help manage the transition to online or telephone sessions. The move away from face-to-face sessions and a greater reliance on technology can also bring a potential risk of feelings such as telepressure [28,29]. With increase in demand and fatigue there is the potential for an increased risk of burnout. As discussed above, the areas of worklife model of burnout considers workload to have a causal link with burnout. As work is now being brought into the home more with increased use of technology there is the potential for workload pressures to increase.

What is unsure at the moment is the level of this potential burnout and what mechanisms can be out in place to support working psychotherapists to enable them to carry on providing important mental health support to others. The issue of burnout in psychotherapists has been looked at over the years with some suggesting that burnout of psychotherapists is an ethical issue [9,30]. Self-care is essential for psychotherapy practice [31], and the British Association of Counselling and Psychotherapy (BACP) have embedded it in their professional code of practice [32].

Taken together, this research considers these concerns and relevant variables that are deemed particularly important during COVID-19. Despite their significant relationships to mental health, as discussed above, those variables have not been evaluated in one single study to appraise the relationships between them. When discussing compassion and psychotherapy practice, the role of the psychotherapists’ self-compassion is often missed. The aim of this study is to further examine the links between self-compassion, burnout, work–life balance and telepressure amongst this under-researched workforce, professional psychotherapists. It is important that the mental health of psychotherapists is prioritised as they play an important role to address COVID-19 mental health difficulties of the public.

### 1.5. Aims

Therefore, this study aimed to explore relationships among burnout, self-compassion, work–life balance and telepressure in professional psychotherapists. Three research questions were established: RQ1.How is burnout related to self-compassion, work–life balance and telepressure?RQ2.RQ2. How is burnout predicted by these variables?RQ3.Does self-compassion moderate the relationship between work–life balance and burnout?

## 2. Materials and Methods

### 2.1. Participants

The inclusion criteria were that participants being 18 years old or older, and a practising psychotherapist, registered with a psychotherapy accreditation body. Participants were recruited via snowball sampling using an online survey distributed in the network of a mental health organisation in the East Midlands region in the United Kingdom in April–May 2020. Of 126 professional psychotherapists who participated to the study, 106 (84%; 83 females, 23 males; Age 47.42 ± 14.00 years old, range 22–75 years old; 96 British, 5 North Americans, 4 other Europeans and 1 Asian) completed four mental health scales, satisfying the required sample size calculated by power analysis (84: two tails, p H1 = 0.30, α = 0.05, Power = 0.80, p H0 = 0; [33]). On average, they had 9.30 years (SD 7.08 years) of practice experience as a registrant. Our sample was similar to the general population of psychotherapists in the UK (male–female ratio and age). Though the reason for withdrawal was not asked to the 20 withdrawn participants per ethical guidelines, no reason nor complaint was received. Ethics approval was obtained from [blind for anonymity].

### 2.2. Instruments

Four scales were used in this study. We chose those four because (i) they were valid and reliable, (ii) they have been used actively in other studies so that comparison would be possible later, and (iii) they were relatively short, minimising the participation workload for busy psychotherapists.

Psychotherapists’ burnout was assessed using the Maslach Burnout Inventory 2 item-version [34] that consisted of one item for emotional exhaustion (‘I feel burned out from my work’) and the other for depersonalisation (‘I have become more callous toward people since I took this job’). These items were responded on a seven-point Likert scale regarding frequency (0 = ‘Never’ to 7 = ‘Every day’).

Self-compassion was evaluated using the Self-Compassion Scale—Short Form (SCS-SF; [35]), a shortened version of the original 26-item Self-Compassion Scale [36]. SCS-SF comprises 12 items on a five-point Likert scale (1 = ’Almost never’ to 5 = ’Almost always’; e.g., ‘I’m intolerant and impatient towards those aspects of my personality I don’t like.’). Cronbach’s alpha was 0.86 [36].

Work–Life Balance Checklist (WLBC) was used to evaluate the level of work–life balance. WLBC consisted of seven items about WLB of psychotherapists (e.g., ‘I have to take work home most evenings.’), responded by A = ‘Agree’, B = ‘Sometimes’, and C = ‘Disagree’ [37]. The response A implies poor WLB, while the response C indicates good WLB. For the purpose of this study, the responses were converted to a five-point Likert scale (1 = ‘Agree’, 3 = ‘Sometimes’, and 5 = ‘Disagree’), hence higher scores indicated better WLB. Cronbach’s alpha was 0.66 [37].

Lastly, telepressure was appraised using the six-item scale responded on a seven-point Likert scale (1 = ‘Strongly Disagree’ to 5 = ‘Strongly Agree’) [29]. The scale asks about message-based technologies that you have control over when to respond (email, text messages, voicemail, etc.) used for your work purposes (e.g., ‘It’s hard for me to focus on other things when I receive a message from someone.’). Cronbach’s alpha was 0.92.

### 2.3. Procedure

First the collected data were screened for outliers and the assumptions of parametric tests. Second, correlations between their burnout, self-compassion, work–life balance and telepressure were evaluated (RQ1). Third, multiple regression analyses were performed to identify significant predictors for burnout (RQ2). Finally, moderation analyses were done to evaluate whether self-compassion would moderate the pathway from work–life balance to burnout (RQ3).

## 3. Results

Analyses were conducted using IBM SPSS version 26.0 and Process Macro [38]. Two scores in depersonalisation were identified as an outlier using the Outlier Labelling Rule [39], so were winsorised [40]. All variables demonstrated good internal reliability (α ≧ 0.85; Table 1).

The mean scores for the two burnout subscales were below the midpoint, whereas those for self-compassion and work–life balance were above. The mean score for telepressure was around the midpoint. The below-midpoint scores in the mental health problem outcomes were similar to previous studies that assessed the mental health of healthcare workers during COVID-19 [41,42].

### 3.1. Relationships among Motivation, Engagement, Resilience, and Self-Criticism/-Compassion (RQ1)

As dedication and amotivation were not normally distributed (Shapiro-Wilk’s test, *p* < 0.05), data were square-root-transformed to satisfy the assumption of normality [43]. Pearson’s correlation was calculated (Table 2).

Both of the burnout subscales—emotional exhaustion and depersonalisation—were positively associated with weekly working hours and telepressure, and negatively associated with age, self-compassion and work–life balance. Emotional exhaustion and depersonalisation were positively associated with each other.

### 3.2. Predictors of Motivation (RQ2)

Multiple regression analyses were conducted to explore the relative contribution of weekly work hours, self-compassion, work–life balance, and telepressure to burnout, namely emotional exhaustion and depersonalisation (Table 3). Gender and age were entered to adjust for their effects (step one), and then weekly work hours, self-compassion, work–life balance, and telepressure were entered (step two). Adjusted coefficient of determination (Adj. R^2^) were reported to appraise the degree of variance in the population. Multicollinearity was of no concern (VIF < 10).

Weekly working hours, self-compassion, work–life balance and telepressure accounted for 38% (large effect size; [39] of the variance in emotional exhaustion, and 14% (medium effect size; [39]) in depersonalisation. Weekly working hours (B = 0.39, *p* < 0.001), work–life balance (B = −0.48, *p* < 0.001) and telepressure (B = 0.24, *p* < 0.008) were significant predictors of emotional exhaustion, and weekly working hours (B = 0.20, *p* < 0.013) and work–life balance (B = −0.36, *p* = 0.022) were significant predictors of depersonalisation. Weekly working hours and work–life balance were significant predictors of both emotional exhaustion and depersonalisation. Work–life balance was the strongest predictor of both burnout subscales: for one unit of increase in work–life balance, emotional exhaustion decreases 0.48 unit, and depersonalisation decreases 0.36 unit.

### 3.3. Mediation of Self-Compassion on Work–Life Balance—Emotional Exhaustion

Path analysis was conducted, using Model 4 in the Process Macro (parallel mediation model; [38]), in order to examine whether self-compassion (mediator variable) mediated the relationship between work–life balance (predictor variable) and emotional exhaustion (outcome variable).

The confidence interval for the indirect effect is a BCa bootstrapped CI based on 5000 samples. Direct effect (total effect). Values attached to arrows are coefficients indicating impacts. * *p* < 0.05, *** *p* < 0.001.

Self-compassion partially mediated the relationship between work–life balance and emotional exhaustion (Figure 1). The total effect of work–life balance on emotional exhaustion, including self-compassion, was significant, b = −0.81, t(103) = −5.45, *p* < 0.001, CI [−1.11, −0.52]. The direct effect of work–life balance on emotional exhaustion was also significant, b = −0.74, t(103) = −4.96, *p* < 0.001, CI [−1.03, −0.44]. The indirect effect of work–life balance on emotional exhaustion, controlling for self-compassion, was not significant, b = −0.08, CI [−0.24, 0.002].

### 3.4. Mediation of Self-Compassion on Work–Life Balance—Depersonalisation

Next another path analysis was conducted to evaluate whether self-compassion (mediator variable) mediated the relationship between work–life balance (predictor variable) and depersonalisation (outcome variable).

The confidence interval for the indirect effect is a BCa bootstrapped CI based on 5000 samples. Direct effect (total effect). Values attached to arrows are coefficients indicating impacts. * *p* < 0.05, *** *p* < 0.001.

Self-compassion did not mediate the relationship between work–life balance and depersonalisation (Figure 2). The total effect of work–life balance on depersonalisation, including self-compassion, was significant, b = −0.56, t(103) = −3.83, *p* = 0.0002, CI [−0.85, −0.27]. The direct effect of work–life balance on depersonalisation was also significant, b = −0.51, t(103) = −3.41, *p* = 0.0009, CI [−0.80, −0.21]. The indirect effect of work–life balance on emotional exhaustion, controlling for self-compassion, was not significant, b = −0.06, CI [−0.22, 0.002].

## 4. Discussion

The mental health of professional psychotherapists has not been widely documented; therefore, little is known about the psychological impact burnout has on this occupational group. Burnout in the healthcare profession has the potential to impair the ability of delivering compassionate patient care. Accordingly, this study evaluated the relationship to which self-compassion, work–life balance and telepressure explain burnout among professional psychotherapists. Burnout was positively associated with telepressure and negatively associated with self-compassion and work–life balance. Work–life balance was the strongest predictor of burnout. Lastly, while self-compassion mediated the pathway from work–life balance to emotional exhaustion, it did not mediate the one from work–life balance to depersonalisation. These findings are discussed in turn below.

### 4.1. Self-Compassion and Burnout

Burnout in professional psychotherapists is negatively associated with self-compassion and work–life balance and positively associated with telepressure (RQ1). In other words, burnout is common in those who have lower self-compassion and work–life balance. Previous empirical studies have shown that the pressure to meet high expectations from ourselves and others can lead to significant burnout [44]. Moreover, burnout is less common in people who have higher levels of self-compassion because they tend to be more resilient to stressful situations [45,46]. Self-compassion serves to protect against the potentially debilitating effects of work-related burnout [47]. Overall, these findings are in accordance with findings reported by Dev, Fernando, Lim, and Consedine (2018), linking higher levels of burnout to lower self-compassion and greater self-compassion with lower burnout. Our study offers an addition to this relationship, explored a sample of psychotherapists.

In line with the present results, previous studies [29,48] have also identified that burnout was positively associated with telepressure. The compulsion to immediately reply to work-related communications and workaholism tends to result in higher levels of burnout and emotional exhaustion [49]. These findings emphasise that maintaining a healthy work–life balance is particularly important for the mental health of psychotherapists, protecting them from burnout. Moreover, previous studies suggest that physical and cognitive exhaustion and sleep quality issues are linked to higher levels of workplace telepressure, highlighting the importance of telepressure in work mental health [48]. The negative relationship between self-compassion and telepressure suggests that cultivating self-compassion may help reduce telepressure. Considering the ever-increasing needs for telemedicine [50], this finding can have impactful implications. Future research should further examine this relationship.

The data presented identifies a clear link between self-compassion and burnout. Recognising how burnout affects psychotherapists’ ability to provide compassionate care may prompt them to seek professional help. Interventions such as compassion-focused therapy or mindful self-compassion, as well as other compassion programmes, may help to reduce burnout and improve self-compassion, but can also raise awareness about the impact burnout has in this specific occupational group [51,52,53].

The relationship between work–life balance and mental health was reported in previous research, and psychological safety was suggested as a key factor that links those two [54]. Our results are similar and may suggest that psychotherapists need to feel safe in the current situation, as emphasised in the framework of self-care among healthcare workers [55]. For example, the British Association for Counselling and Psychotherapy has been offering support among professional psychotherapists through virtual gatherings, etc. Socialisation has been used in other caring professions to maintain wellbeing [56]. Active use of those measures may help them feel safe, leading to better mental health, hence lower burnout. However, a caution is needed as some psychotherapists may feel guilt towards taking care for themselves [55]. While offering support to facilitate work–life balance, addressing the self-care guilt in psychotherapists would be helpful. Future research is needed to evaluate the impacts of those activities.

### 4.2. Predictors of Burnout

The results of our regression analyses are consistent with other research [29,48], which found that burnout was positively predicted by telepressure and negatively predicted by self-compassion and work–life balance (RQ2). Analysis of potential burnout predictors in our study determined that heightened emotional exhaustion was attributed to long working hours. Work–life balance was the strongest predictor of both emotional exhaustion and depersonalisation among psychotherapists. Previous research demonstrates that other caring occupations such as nurses [57], physicians [58], and teachers [59] who work long hours also experience emotional exhaustion. Our findings, in addition to those previous ones, suggest that a healthy work–life balance and self-compassion can help psychotherapists avoid burnout. While increased levels of self-compassion are linked to lower levels of burnout [51], developing self-compassion would prove effective in mitigating emotional exhaustion.

The areas of worklife model provides a structured framework for considering areas of work-life as predictors of burnout [60]. The findings have implications for future research on burnout [61]. It is important to raise awareness of workaholism and the “always connected” attitude in the workplace, which frequently blurs the lines between home life and work-life. This study, in particular, identifies the need to address workplace telepressure as well as the demand or expectation that employees be available and accessible at all times [48]. Recommendations for future research could include the investigation of other potential risk factors of burnout in psychotherapists, such as the level to which burnout across job settings is affected; that is, whether working in a private setting could help to reduce feelings of stress, emotional exhaustion, depersonalisation and burnout [61].

### 4.3. Mediation of Self-Compassion on Work–Life Balance

Based on the parallel mediation model, we confirmed that self-compassion mediated the pathway between work–life balance and emotional exhaustion but did not mediate the relationship between work–life balance and depersonalisation (RQ3). These findings support previous studies [62], indicating that self-compassion is linked to a better work–life balance and a lower risk of burnout.

One possibility may be that self-compassion is more relevant to emotion regulation, as described in the three emotion regulatory systems. Of the two subscales of burnout, emotion exhaustion captures the emotional aspect of burnout, whereas depersonalisation focuses more on the cognitive aspect of burnout, i.e., impaired and distorted perception of oneself, others and environment. Our results may imply that such cognitive change may not come together with self-compassion (e.g., emotion-cognition mismatch). Alternative interpretation may be that one component of self-compassion is mindfulness, and one salient weakness of mindfulness is depersonalisation. While self-compassion may help them be mindful, it can lead psychotherapists to detach from the experience. Burned out psychotherapists who experience depersonalisation may not be helped using self-compassion. Indeed, evidence in treatment for depersonalisation still remains to be accumulated [63]. As depersonalisation falls in the cognitive domain, mental health approaches such as CBT are suggested [64]. On the other hand, psychotherapists who are emotionally exhausted may benefit from practising self-compassion. Self-compassion interventions have begun to be utilised actively in healthcare, and the promising effects have been reported [65]. How psychotherapists would benefit from these interventions needs to be evaluated in the future.

### 4.4. Limitations

While this research provides valuable insight into psychotherapist burnout, it is important to keep the study’s limitations in mind when interpreting the results. First, our sample was recruited from one UK organisation, which limits the generalisability of our findings. Likewise, the online survey might have discouraged some psychotherapists to participate, as opposed to a hard-copy one. A more representative sample should be used for future research. In addition, self-compassion was measured using the Self-Compassion Scale—Short Form [35]; however, there is doubt as to its validity [66]. Further, our variables were measured using self-reported scales, which restricts the reliability of participant responses [67]. Individuals are often biased when reporting their own experiences, as self-reported surveys may influence results. In other words, respondents are more likely to report socially acceptable or desirable experiences [68]. Moreover, this study assessed the mental health during COVID-19, pre-existing mental health symptoms in psychotherapists were not focused on [69,70]. As such, future research could look at alternative measures for collecting data. Lastly, our study did not explore causality of these variables. Future research should elucidate causal directions of these relationships using research designs such as longitudinal studies.

## 5. Conclusions

This study examined burnout in professional psychotherapists, and its relationships with self-compassion, work–life balance and telepressure. Burnout was positively correlated with weekly working hours and telepressure, and negatively correlated with age, self-compassion and work–life balance. Weekly working hours and work–life balance were significant predictors. Moreover, self-compassion partially mediated the relationship between work–life balance and emotional exhaustion. Our findings suggest that supporting work–life balance and self-compassion would be particularly helpful to prevent psychotherapists feeling burned out. Self-care training and virtual gatherings may be helpful for them to be involved. Managers and professional bodies should consider these measures to maintain this critical workforce during the pandemic. Future research should evaluate these relationships more thoroughly implementing longitudinal and/or interventional studies.

## Figures and Tables

**Figure 1 ijerph-18-05308-f001:**
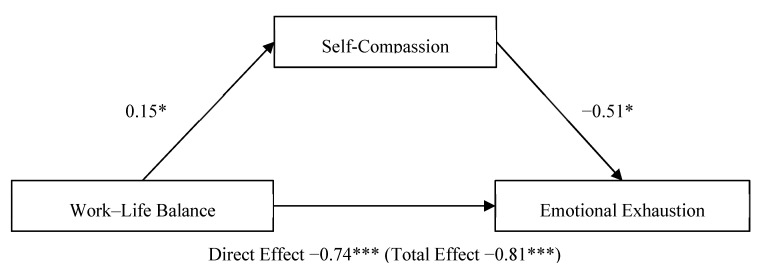
Parallel mediation model: work–life balance as a predictor of emotional exhaustion, mediated by self-compassion. * *p* < 0.05; *** *p* < 0.001.

**Figure 2 ijerph-18-05308-f002:**
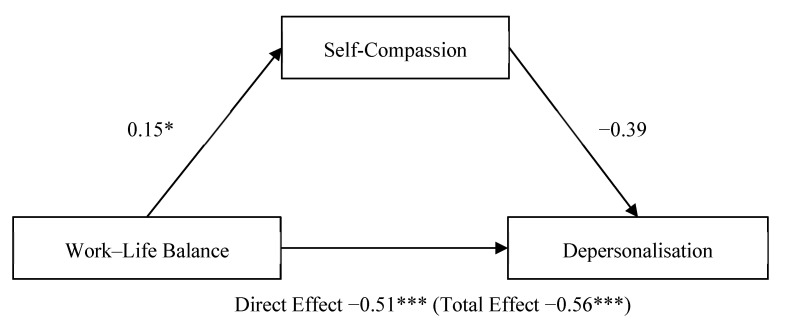
Parallel mediation model: work–life balance as a predictor of depersonalisation, mediated by self-compassion. * *p* < 0.05; *** *p* < 0.001.

**Table 1 ijerph-18-05308-t001:** Descriptive statistics: burnout, self-compassion, work–life balance and telepressure in professional psychotherapists (*n* = 106).

Scale	Constructs (Range)	M	SD	α
Maslach Burnout Inventory 2-Item Version	Emotional Exhaustion (1–7)	2.75	1.66	NA
	Depersonalisation (1–7)	1.86	1.48	NA
Self-Compassion Scale-Short Form	Self-Compassion (1–5)	3.67	0.73	0.90
Work–Life Balance Checklist	Work–Life Balance (1–5)	3.37	0.96	0.85
6-Item Telepressure Scale	Telepressure (1–5)	2.58	0.97	0.93

**Table 2 ijerph-18-05308-t002:** Correlations among burnout, self-compassion, work–life balance and telepressure in professional psychotherapists (*n* = 106).

		1	2	3	4	5	6	7	8	9
1	Gender (0 = F, M = 1)	-								
2	Age	0.13	-							
3	Experience	0.02	0.60 **	-						
4	Weekly Working Hours	0.19	−0.02	0.32 **	-					
5	Emotion Exhaustion	−0.05	−0.41 **	−0.01	0.48 **	-				
6	Depersonalisation	0.13	−0.27 **	−0.08	0.31 **	0.57 **	-			
7	Self-Compassion	0.04	0.40 **	0.34 **	0.03	−0.30 **	−0.24 *	-		
8	Work–Life Balance	−0.07	0.04	0.04	−0.30 **	−0.47 **	−0.35 **	0.21 *	-	
9	Telepressure	−0.05	−0.25 **	−0.19	−0.15	0.29 **	0.20 *	−0.29 **	−0.19	-

* *p* < 0.05, ** *p* < 0.01.

**Table 3 ijerph-18-05308-t003:** Multiple regression: weekly working hours, self-compassion, work–life balance and telepressure to burnout among professional psychotherapists (*n* = 106).

	Emotional Exhaustion			Depersonalisation		
		95% CI			95% CI	
	B	Lower	Upper	B	Lower	Upper
Step 1						
Gender (0 = F, 1 = M)	0.004	−0.20	0.21	0.17	−0.02	0.37
Age	−0.19 ***	−0.27	−0.17	−0.12 **	−0.20	−0.04
Step 2						
Gender (0 = F, 1 = M)	−0.12	−0.27	0.04	0.11	−0.08	0.29
Age	−0.14 ***	−0.21	−0.07	−0.09 *	−0.17	−0.004
Weekly Working Hours	0.39 ***	0.26	0.52	0.20 *	0.04	0.35
Self-Compassion	−0.19	−0.56	0.18	−0.20	−0.63	0.24
Work–Life Balance	−0.48 ***	−0.73	−0.23	−0.36 *	−0.66	−0.05
Telepressure	0.24 **	0.06	0.41	0.13	−0.07	0.34
Adj. R^2^ Δ		38%			14%	

B = unstandardised regression coefficient. * *p* < 0.05; ** *p* < 0.01; *** *p* < 0.001.

## Data Availability

The data presented in this study are available on request from the corresponding author. The data are not publicly available due to ethical restrictions.

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
