# Peer review of "Burnout in Professional Psychotherapists: Relationships with Self-Compassion, Work–Life Balance, and Telepressure"

_ijerph, 2021, doi:10.3390/ijerph18105308_

Round 1
Reviewer 1 Report
The premises of the study are very interesting and relevant, since researchers focused on mental health of a professional category - professional psychologists - that has been on the Covid19 forefront without receiving an in-depth evaluation.
However, the paper need some improvement.
- bibliography and introduction section should be expanded and authors should better discuss the choice to deepen precisely the constructs of self-compassion, work-live balance, telepressure
- authors should make explicit the reasons for the choice of scales
- authors should provide a wider overview on consequences and practical implications of their findings
Author Response
Dear Reviewer 1
Thank you for your helpful feedback. We have systematically revised our manuscript addressing the points you have raised. We hope this revised paper is now acceptable for publication. We extend our sincere gratitude to you for your feedback that has significantly helped to strengthen the paper.
Reviewer 1’s comment 1 (1-1)
The premises of the study are very interesting and relevant, since researchers focused on mental health of a professional category - professional psychologists - that has been on the Covid19 forefront without receiving an in-depth evaluation.
However, the paper need some improvement.
bibliography and introduction section should be expanded and authors should better discuss the choice to deepen precisely the constructs of self-compassion, work-live balance, telepressure
Authors’ response 1-1
Thank you for your kind words, and helpful suggestions. In line with your comment, now a justification for the choice of variables is added, and relatedly, the relevance to COVID-19 for each variable is clarified.
Reviewer 1’s comment 2 (1-2)
authors should make explicit the reasons for the choice of scales
Authors’ response 1-2
In line with your comment, the reason for the choice of scales is now clarified.
Reviewer 1’s comment 3 (1-3)
authors should provide a wider overview on consequences and practical implications of their findings
Authors’ response 1-3
In line with your comment, the discussion section is now strengthened by providing a wider overview. Thank you.
Reviewer 2 Report
This topic is very interesting and current, reporting the role of the psychotherapists' self-compassion. Please look at these points:
- Lines 32-33 "One area that appears to have... research being hospital-based staff" This sentence did not sound good from a grammatical point of view, try to change it.
- Lines 35-39: "With up to 40% of healthcare to mental health.. in need". Please stress this important concept. Look at this ref. Understanding the Impact of Initial COVID-19 Restrictions on Physical Activity, Wellbeing and Quality of Life in Shielding Adults with End-Stage Renal Disease in the United Kingdom Dialysing at Home versus In-Centre and Their Experiences with Telemedicine. Int J Environ Res Public Health. 2021 Mar 18;18(6):3144. doi: 10.3390/ijerph18063144.
- Lines 80-83: "Increased workload and associated pressure impact work-life balance, which is a contributing factor to burnout." This concept is probably the most important of the paper. Please highlight it. Occupational burnout syndrome and post-traumatic stress among healthcare professionals during the novel coronavirus disease 2019 (COVID-19) pandemic. Best Pract Res Clin Anaesthesiol. 2020 Sep;34(3):553-560.
- Lines 103-104: "What is unsure at the moment is the.. providing important mental health support to others." The concept of telepressure is closely associated with telemedicine. Please look at these ref. Will COVID-19 change neurosurgical clinical practice? Br J Neurosurg. 2020 Jun 1:1-2. doi: 10.1080/02688697.2020.1773399.
- Lines 121-127: What do you mean for H2, H3, RQ1? Please explain or remove them.
- Table 1 is very important and should be discuss more in the text in the results section. Please look at these studies XiaoPsychological impact of healthcare workers in China during COVID-19 pneumonia epidemic: A multi-center cross-sectional survey investigation. J Affect Disord. 2020 Sep 1;274:405-410. doi: 10.1016/j.jad.2020.05.081 - Mental Health and Health-Related Quality-of-Life Outcomes Among Frontline Health Workers During the Peak of COVID-19 Outbreak in Vietnam: A Cross-Sectional Study. Risk Manag Healthc Policy. 2020
- Figure 1 and figure 2 seem very similar, please discuss more in the text.
- Lines 275: " In other words, burnout is common in those who have lower... from ourselves and others can lead to significant burnout." Please underline that Covid made us forget old psychiatric and medical disease. Look at these ref: Intracranial hemorrhage and COVID-19, but please do not forget "old diseases" and elective surgery. Brain Behav Immun. 2021 Feb;92:207-208. doi: 10.1016/j.bbi.2020.11.034. - Stress, Anxiety, and Depression in People Aged Over 60 in the COVID-19 Outbreak in a Sample Collected in Northern Spain. Am J Geriatr Psychiatry. 2020 Sep;28(9):993-998. doi: 10.1016/j.jagp.2020.05.02
- Lines 317-320: ". Previous research demonstrates that other occupations such as nurses [52], physicians [53], and teachers.. These findings suggest that a healthy work-life balance and self-compassion can help psychotherapists avoid burnout. " what is the correlation between these two sentences?
- This study probably has some limitations: for example the sample is completely random, however as it was online survey, it probably that older psychotherapist decided not to take part. Please add few lines at the end of the discussion section.
Overall a good paper.
Author Response
Dear Reviewer 2
Thank you for your helpful feedback. We have systematically revised our manuscript addressing the points you have raised. We hope this revised paper is now acceptable for publication. We extend our sincere gratitude to you for your feedback that has significantly helped to strengthen the paper.
Reviewer 2’s comment 1 (2-1)
This topic is very interesting and current, reporting the role of the psychotherapists' self-compassion. Please look at these points:
Lines 32-33 "One area that appears to have... research being hospital-based staff" This sentence did not sound good from a grammatical point of view, try to change it.
Authors’ response 2-1
Thank you for your kind words. Now the Lines 32-33 are amended: “... with the main focus of research being hospital-based staff” is removed, and adjustment is made in the next sentence.
Reviewer 2’s comment 2 (2-2)
Lines 35-39: "With up to 40% of healthcare to mental health.. in need". Please stress this important concept. Look at this ref. Understanding the Impact of Initial COVID-19 Restrictions on Physical Activity, Wellbeing and Quality of Life in Shielding Adults with End-Stage Renal Disease in the United Kingdom Dialysing at Home versus In-Centre and Their Experiences with Telemedicine. Int J Environ Res Public Health. 2021 Mar 18;18(6):3144. doi: 10.3390/ijerph18063144.
Authors’ response 2-2
In line with your comment, this sentence is now revised. We have looked at the interesting paper you suggested, and used it in the discussion, where we talk about the relationship between self-compassion and telepressure.
However in order to further stress this concept, this paper is added;
https://bmcpublichealth.biomedcentral.com/articles/10.1186/s12889-020-09322-z
Reviewer 2’s comment 3 (2-3)
Lines 80-83: "Increased workload and associated pressure impact work-life balance, which is a contributing factor to burnout." This concept is probably the most important of the paper. Please highlight it. Occupational burnout syndrome and post-traumatic stress among healthcare professionals during the novel coronavirus disease 2019 (COVID-19) pandemic. Best Pract Res Clin Anaesthesiol. 2020 Sep;34(3):553-560.
Authors’ response 2-3
In line with your suggestion, this sentence is now strengthened using the findings from the paper suggested.
Reviewer 2’s comment 4 (2-4)
Lines 103-104: "What is unsure at the moment is the.. providing important mental health support to others." The concept of telepressure is closely associated with telemedicine. Please look at these ref. Will COVID-19 change neurosurgical clinical practice? Br J Neurosurg. 2020 Jun 1:1-2. doi: 10.1080/02688697.2020.1773399.
Authors’ response 2-4
Thank you again for your helpful suggestion. In line with your comment, the suggested article is inserted to a few sentences before as the paragraph discusses telepressure; “The move away from face-to-face sessions and a greater reliance on technology can also bring a potential risk of feelings such as telepressure“
Reviewer 2’s comment 5 (2-5)
Lines 121-127: What do you mean for H2, H3, RQ1? Please explain or remove them.
Authors’ response 2-5
Sorry about this. Now those are corrected: RQ1, RQ2 and RQ3.
Reviewer 2’s comment 6 (2-6)
Table 1 is very important and should be discuss more in the text in the results section. Please look at these studies XiaoPsychological impact of healthcare workers in China during COVID-19 pneumonia epidemic: A multi-center cross-sectional survey investigation. J Affect Disord. 2020 Sep 1;274:405-410. doi: 10.1016/j.jad.2020.05.081 - Mental Health and Health-Related Quality-of-Life Outcomes Among Frontline Health Workers During the Peak of COVID-19 Outbreak in Vietnam: A Cross-Sectional Study. Risk Manag Healthc Policy. 2020
Authors’ response 2-6
In line with your comment, a description of Table 1 is now added, using the informative papers you have suggested. Thank you.
Reviewer 2’s comment 7 (2-7)
Figure 1 and figure 2 seem very similar, please discuss more in the text.
Authors’ response 2-7
In line with your comment, the difference between the two figures (Figure 1 evaluated emotional exhaustion, while Figure 2 evaluated depersonalisation) are now italicised in the description to highlight the difference.
Reviewer 2’s comment 8 (2-8)
Lines 275: " In other words, burnout is common in those who have lower... from ourselves and others can lead to significant burnout." Please underline that Covid made us forget old psychiatric and medical disease. Look at these ref: Intracranial hemorrhage and COVID-19, but please do not forget "old diseases" and elective surgery. Brain Behav Immun. 2021 Feb;92:207-208. doi: 10.1016/j.bbi.2020.11.034. - Stress, Anxiety, and Depression in People Aged Over 60 in the COVID-19 Outbreak in a Sample Collected in Northern Spain. Am J Geriatr Psychiatry. 2020 Sep;28(9):993-998. doi: 10.1016/j.jagp.2020.05.02
Authors’ response 2-8
Thank you for your valuable input. These papers are more relevant to the limitations of our study, therefore added to the limitation section.
Reviewer 2’s comment 9 (2-9)
Lines 317-320: ". Previous research demonstrates that other occupations such as nurses [52], physicians [53], and teachers.. These findings suggest that a healthy work-life balance and self-compassion can help psychotherapists avoid burnout. " what is the correlation between these two sentences?
Authors’ response 2-9
In line with your comment, these sentences are now revised to enhance the clarify and coherence.
Reviewer 2’s comment 10 (2-10)
This study probably has some limitations: for example the sample is completely random, however as it was online survey, it probably that older psychotherapist decided not to take part. Please add few lines at the end of the discussion section.
Overall a good paper.
Authors’ response 2-10
In line with your comment, it is now added to the limitations.
Round 2
Reviewer 1 Report
I have no more remark on the paper
Reviewer 2 Report
Authors solved all my criticisms.